# Communicating Arteries and Leptomeningeal Collaterals: A Synergistic but Independent Effect on Patient Outcomes after Stroke

**Sara Sablić** [1], **Krešimir Dolić** [1,2,3], **Danijela Budimir Mršić** [1,2], **Mate Čičmir-Vestić** [4], **Antonela Matana** [3], **Sanja Lovrić Kojundžić** [1,2,3] and **Maja Marinović Guić** [1,2,3,*]

1   Clinical Department of Diagnostic and Interventional Radiology, University Hospital of Split, 21000 Split, Croatia; sarasablic@gmail.com (S.S.); kdolic79@gmail.com (K.D.); danijelabudimir@gmail.com (D.B.M.); lovric.sanja@gmail.com (S.L.K.)
2   School of Medicine, University of Split, 21000 Split, Croatia
3   University Department of Health Studies, University of Split, 21000 Split, Croatia; antonela.matana@gmail.com
4   Department of Neurology, University Hospital of Split, 21000 Split, Croatia; mate.c-v@hotmail.com
*   Correspondence: maja.marinovic.guic@gmail.com

**Abstract:** The collateral system is a compensatory mechanism activated in the acute phase of an ischemic stroke. It increases brain perfusion to the hypoperfused area. Arteries of the Willis' circle supply antegrade blood flow, while pial (leptomeningeal) arteries direct blood via retrograde flow. The aim of our retrospective study was to investigate the relationship between both collateral systems, computed tomography perfusion (CTP) values, and functional outcomes in acute stroke patients. Overall, 158 patients with anterior circulation stroke who underwent mechanical thrombectomy were included in the study. We analyzed the presence of communicating arteries and leptomeningeal arteries on computed tomography angiography. Patients were divided into three groups according to their collateral status. The main outcomes were the rate of functional independence 3 months after stroke (modified Rankin scale score, mRS) and mortality rate. Our study suggests that the collateral status, as indicated by the three groups (unfavorable, intermediate, and favorable), is linked to CT perfusion parameters, potential recuperation ratio, and stroke outcomes. Patients with favorable collateral status exhibited smaller core infarct and penumbra volumes, higher mismatch ratios, better potential for recuperation, and improved functional outcomes compared to patients with unfavorable or intermediate collateral status.

**Keywords:** cerebral infarction; stroke; mechanical thrombectomy; endovascular treatment; collateral circulation

## 1. Introduction

Acute ischemic stroke is a medical condition that poses a threat to life and disability [1]. Numerous advances have been made in research on its pathogenesis and treatment. Activation of the collateral system has been extensively studied in the past few years as a contributing factor to a better prognosis following stroke [2–7], regardless of whether intravenous thrombolytic agent administration was the primary treatment or if it was followed by endovascular treatment (EVT) [8–12]. While mechanical thrombectomy (MT) is unquestionably an effective therapy approach, it is not always associated with favorable functional outcomes. The time window for MT was extended from 6 h to even 24 h from the onset of the stroke due to the great results of this treatment method [13–15]. However, some patients do not benefit from this treatment, even when the EVT is administered rapidly after stroke onset. Several factors contribute to this, such as older age, diabetes, a higher NIHSS score, an occlusion site (internal carotid artery occlusion), a lower ASPECT score, and distal clot migration [16–18]. The term "fast progressors" refers to patients

who quickly deteriorate due to fast infarct progression [19,20]. These patients usually have terminal carotid occlusions and impaired collateral flow via the anterior or posterior cerebral artery, which indicates the importance of collateral rerouting of blood flow [19,20]. Numerous studies that examined pial arteries using CT angiography (CTA) discovered that a healthy collateral system "saves" the hypoperfused area of the brain and can even extend the time window for potential mechanical thrombectomy treatment [21]. Arteries of the Willis circle (mainly communicating arteries) as branches of the internal carotid artery are considered primary collaterals. They abruptly redirect blood flow to the site of occlusion. Small arterial branches and leptomeningeal arteries (branches of the external carotid artery) are considered secondary collaterals, and they retrogradely supply branches of the occluded artery [22,23]. Few studies have evaluated anatomic variants of the Willis' circle (CoW), while other studies analyzed secondary collateral scores on CT angiography and their impact on outcomes after ischemic stroke [9,10,24–27]. These studies concluded that patients with better collateral status had a smaller computed tomography perfusion (CTP) core and penumbra, a larger mismatch ratio, and better functional outcomes. In addition, baseline collateral status is significantly associated with clinical and safety outcomes in acute ischemic stroke patients treated with reperfusion therapy [28]. The most recent guidelines from the American Heart Association/American Stroke Association indicate that it would be appropriate to take collateral flow status into account when making clinical decisions about which individuals are eligible for mechanical thrombectomy [29].

The aim of our study was to make a comprehensive analysis of both collateral systems on CTA and investigate the relationship between collateral status, CT perfusion parameters, and functional outcomes in anterior circulation acute ischemic stroke patients treated with endovascular therapy.

## 2. Materials and Methods

### 2.1. Study Design and Participants

This study complied with the ethical guidelines of the 1975 Declaration of Helsinki and was approved by the institutional ethics committee at the University Hospital in Split, Croatia (protocol code 2181-147-01/06/LJ.Z.-23.02). Informed consent was waived due to the retrospective nature of the study. We examined patients with acute ischemic stroke in the anterior circulation who were eligible for endovascular therapy at our comprehensive stroke center from 1 January 2021 to 1 July 2023. The inclusion criteria included the following: patients aged 18 years and older, onset of neurological symptoms less than six hours before presentation to the Neurology Emergency Department, and radiological confirmation of the occlusion. Exclusion criteria included the following: patients with acute ischemic stroke in the posterior circulation (vertebral or basilar artery) and patients with stroke caused by carotid artery dissection. Our inclusion criteria were met by 158 patients. We gathered clinical data (National Institutes of Health Stroke Scale (NIHSS) on admission, onset-to-door time, door-to-needle time, and onset-to-groin puncture time) as well as patient demographics (age, sex, stroke risk factors, comorbidities, and anticoagulant treatment) from the hospital information system (HIS).

### 2.2. Radiological Data

Prior to endovascular treatment, every patient underwent a standardized stroke protocol that includes emergency nonenhanced brain CT, cerebral CT angiography, and CT perfusion on a 128-slice MSCT Siemens Somatom Definition AS, Erlangen, Germany. All scans were analyzed using Syngovia software (version VB60A, Hofix08). Cerebral arteries were evaluated using the advanced vessel analysis software (syngo.via CT vascular, version VB60A, Hofix08), while perfusion analysis was performed in syngo.via Perfusion CT Neuro software (version VB60A, Hofix08). Two experienced neuroradiologists evaluated cerebral CT angiography to determine the presence of different anatomical variants of CoW. The presence of anterior and posterior communicating arteries was observed (present/absent) without measuring their diameters. Primary collaterals were evaluated

as described: Patients were categorized as absent (no communicating arteries present) or present (anterior and/or posterior communicating arteries). Anterior communicating artery (AcoA) was considered positive only if an ipsilateral A1 segment was present on the side of the occluded vessel.

If the patient had ACoA but aplasia of the A1 segment on the ipsilateral side, ACoA was considered absent. Present posterior communicating artery (PCoA) was considered positive only if it was present on the side of the occluded vessel.

The collateral grade score of secondary collaterals was evaluated on CT angiography with the collateral score proposed by Tan et al. [30], which uses a 4-point grading system (0–3) and grades vessel filling in the territory of the occluded artery to assess collateral circulation. A Tan score of 0 indicates absent filling, with no visible vessels within the occluded middle cerebral artery (MCA) territory. A Tan score of 1 indicates arterial contrast filling of $\leq 50\%$ of the occluded MCA territory. A score of 2 was given when contrast filling was present in >50% but <100% of the occluded MCA territory, and a score of 3 was given when contrast filling was present in 100% of the occluded MCA territory. Low Tan scores range from 0 to 1, while high Tan scores range from 2 to 3.

### 2.3. Patient Grouping

We categorized patients into three groups: Group 1 (unfavorable collateral group) consisted of patients who had absent communicating arteries and a low Tan score (0 or 1); group 2 (intermediate collateral group) consisted of patients who had at least one communicating artery present and a low Tan score or expressed a high Tan score (2 or 3) but had absent communicating arteries; group 3 (favorable collateral group) consisted of patients who had at least one communicating artery and who had a high Tan score (2 or 3).

### 2.4. Patient Management

Mechanical thrombectomy was performed under local or general anesthesia with approved endovascular devices (aspiration catheters, stent retrievers, or a combination of both).

### 2.5. Outcomes and Measures

The main outcome was the rate of functional independence measured by the modified Rankin scale score (0–6) 90 days after a stroke incident. The favorable mRS score was considered 0–2.

Additional outcomes that were automatically calculated by software were volumes of core infarct and penumbra on CT perfusion, mismatch ratio (penumbra divided by the infarct core volume), and the potential recuperation ratio, which is calculated using a formula [31]. The recanalization grade on mechanical thrombectomy was scored using the TICI scale [32], and TICI scores 2B, 2C, and 3 were considered a successful recanalization. We further evaluated the hypodense area on follow-up unenhanced brain CT scans that were obtained approximately 24 h after the endovascular treatment. A score of 0 was assigned to patients who had no infarction detected; a score of 1 included patients with small/lacunar ischemic lesion; a score of 2 was assigned to patients with larger ischemic lesions that occupied less than 50% of the vascular territory of the occluded vessel; and a score of 3 included patients with ischemic lesions that occupied more than 50% of the vascular territory of the affected vessel.

### 2.6. Statistical Analysis

For normality checking, the Kolmogorov–Smirnov test was used. Continuous variables with normal distribution were presented as mean (standard deviation, SD); non-normal variables were reported as median (interquartile range, IQR). Categorical variables are expressed with frequencies (percentages). Differences in categorical variables were analyzed using Pearson chi-square test. Kruskal–Wallis test was used for not normally distributed continuous variables, while ANOVA was used for normally distributed con-

tinuous variables. Statistically significance was considered for two-sided *p*-values less than 0.05. Statistical analysis was conducted using Statistical Package Software for Social Science, version 28 (SPSS Inc., Chicago, IL, USA).

## 3. Results

This single-center retrospective study included 158 patients with acute ischemic stroke in the anterior circulation. All groups had comparable age and sex distribution. The unfavorable collateral group had the highest incidence of arterial hypertension ($p = 0.023$) and atrial fibrillation ($p = 0.029$). The favorable collateral group had the lowest NIHSS median on admittance ($p = 0.011$). Systolic and diastolic blood pressure values were similar among groups. All three groups had comparable onset-to-door, door-to-needle, door-to-puncture, and onset-to-groin puncture times (Table 1). Twenty-seven patients had wake-up strokes, or the onset of the neurological deficit was unknown. Groups did not differ in thrombus localization ($p = 0.176$), and the most common site of occlusion was the M1 segment of the middle cerebral artery (Table 1).

**Table 1.** Baseline characteristics.

| | Unfavorable Collateral Group (n = 24) | Intermediate Collateral Group (n = 69) | Favorable Collateral Group (n = 65) | *p*-Value |
|---|---|---|---|---|
| Age, years (median, IQR) | 82.5 (11.5) | 78 (17) | 75 (18) | 0.051 * |
| Male sex (n, %) | 13 (54.2%) | 46 (66.7%) | 39 (60%) | 0.503 † |
| Arterial hypertension (n,%) | 22 (91.7%) | 49 (72.1%) | 39 (61.9%) | 0.023 † |
| Diabetes mellitus (n,%) | 5 (20.8%) | 15 (22.1%) | 15 (23.8%) | 0.948 † |
| Stroke cause<br>Atrial fibrillation (n,%)<br>Atherosclerosis (n,%) | 15 (65.2%)<br>8 (34.8%) | 43 (63.2%)<br>25 (36.8%) | 26 (41.9%)<br>36 (58.1%) | 0.029 † |
| Anticoagulant/antithrombotic therapy (n,%) | 13 (54.2%) | 28 (41.2%) | 19 (30.2%) | 0.104 † |
| NIHSS on admittance, median (IQR) | 15 (3) | 15 (3.8) | 12 (6.5) | 0.011 * |
| Systolic blood pressure (mmHg, median IQR) | 150 (53.5) | 140 (40) | 150 (45) | 0.952 * |
| Diastolic blood pressure (mmHg, median IQR) | 80 (28.8) | 80 (20) | 80 (20) | 0.988 * |
| Onset-to-door (minutes, median IQR) | 87.5 (101.5) | 84 (74) | 80 (47) | 0.747 * |
| Door-to-needle (minutes, median IQR) | 101.14 (29.8) | 113.9 (38.4) | 107.5 (29.3) | 0.277 ‡ |
| Onset-to-puncture (minutes, median IQR) | 189 (84) | 189 (88) | 191 (59) | 0.833 * |
| Intravenous thrombolysis (n,%) | 9 (37.5%) | 30 (43.5%) | 24 (36.9%) | 0.717 † |
| Thrombus localization<br>MCA occlusion<br>ICA + MCA occlusion | 16 (66.7%)<br>8 (33.3%) | 54 (78.2%)<br>15 (21.7%) | 55 (84.6%)<br>10 (15.4%) | 0.176 † |

* Kruskal–Wallis test. † chi-square test. ‡ ANOVA. NIHSS, The National Institutes of Health Stroke Scale; MCA, middle cerebral artery; ICA, internal carotid artery.

Patients in the favorable collateral group had the lowest volume of core infarct ($p = 0.001$) and penumbra on CT perfusion maps ($p = 0.033$) (Table 2). The mismatch ratio increased with a more developed collateral system and was the highest in the favorable collateral group ($p \leq 0.001$). During mechanical thrombectomy procedures, similar devices were used in all groups ($p = 0.915$); only the aspiration catheters were used in

most patients. Unsuccessful placement of endovascular devices was similar in all groups, predominantly because of anatomic or groin puncture difficulties. The potential recuperation ratio was the lowest in patients with poorly developed collaterals, with the highest percentage in patients with developed both primary and secondary collaterals ($p \leq 0.001$).

**Table 2.** CT perfusion analysis and stroke outcomes.

| | Unfavorable Collateral Group (n = 24) | Intermediate Collateral Group (n = 69) | Favorable Collateral Group (n = 65) | *p*-Value |
|---|---|---|---|---|
| Core infarct, cm³ (median, IQR) | 25.75 (27.9) | 23.11 (29.2) | 11.64 (12.1) | 0.001 * |
| Penumbra, cm³ (median, IQR) | 49.69 (45.1) | 54.74 (39.3) | 39.69 (27.2) | 0.033 * |
| Mismatch ratio (median, IQR) | 1.72 (1.1) | 2.3 (1.7) | 3.55 (3.2) | <0.001 * |
| PRR% (median, IQR) | 66.8 (10.6) | 70.52 (9.3) | 77.57 (10.3) | <0.001 † |
| Mechanical thrombectomy technique:<br>Aspiration only<br>Aspiration catheter + stent retriever<br>Unsuccessful placement | <br>17 (70.8%)<br>2 (33.3%)<br>5 (20.8%) | <br>46 (66.7%)<br>11 (15.9%)<br>12 (17.4%) | <br>44 (67.7%)<br>10 (15.4%)<br>11 (16.9%) | <br><br>0.915 ‡ |
| Successful recanalization (n,%) | 7 (29.2%) | 36 (53.7%) | 32 (49.2%) | 0.219 ‡ |
| TICI 3 (n,%) | 11 (45.8%) | 40 (59.7%) | 39 (59%) | 0.476 ‡ |
| Number of passes (median, IQR) | 2 (2) | 1 (2) | 1 (1.8) | 0.430 * |
| First-pass recanalization (n,%) | 7 (36.8%) | 36 (59%) | 32 (57.1%) | 0.219 ‡ |
| Control CT area of ischemia<br>None<br>Lacunar<br><50%<br>>50% | <br>1 (47%)<br>6 (25%)<br>10 (41.7%)<br>7 (29.2%) | <br>9 (13.2%)<br>23 (33.8%)<br>20 (29.4%)<br>16 (23.5%) | <br>10 (15.4%)<br>32 (49.2%)<br>16 (24.6%)<br>7 (10.7%) | <br>0.366 ‡<br>0.056 ‡<br>0.291 ‡<br>0.072 ‡ |
| SAH or ICH (n,%) | 7 (29.2%) | 13 (19.1%) | 16 (24.6%) | 0.551 ‡ |
| Favorable mRS score (0–2) (n,%) | 4 (17.4%) | 24 (44.4%) | 31 (60.8%) | 0.002 ‡ |
| Death (n,%) | 14 (58.3%) | 23 (37.7%) | 12 (20.3%) | 0.003 ‡ |

* Kruskal–Wallis test. † ANOVA. ‡ chi-square test. PRR, potential recuperation ratio; TICI, thrombolysis in cerebral infarction scale; SAH, subarachnoid hemorrhage; ICH; intracerebral hemorrhage; mRS, modified Rankin scale score.

Successful recanalization (TICI 2B-C) and first-pass recanalization rates were achieved similarly in all groups. We found no difference in the incidence of postprocedural complications among groups ($p = 0.551$) (Table 2). The rate of good functional recovery increased among groups; patients with poor collateral activation had the lowest percentage of mRS 0–2 ($p = 0.002$) and the highest mortality rate ($p = 0.003$). Patients with the most collaterals present had the best functional outcomes (Table 2).

We further evaluated if the presence of primary collaterals affected recruiting secondary collaterals, and we found no statistical significance ($p = 0.313$). A good functional outcome was associated with good primary collaterals ($p = 0.04$) as well as a good TAN score on CT angiography ($p = 0.001$).

## 4. Discussion

Our study findings highlight the influence of collateral circulation on patients' prognosis following an acute ischemic stroke. Our key finding is that acute stroke patients who have a more developed collateral system (a more closed circle of Willis and well-activated leptomeningeal arteries) have better functional recovery and experience a lower mortality rate.

We assessed the effects of primary and secondary collaterals on the radiological and clinical outcomes of patients by proposing a classification that covered both types of collaterals. We analyzed whether the communicating arteries (both ACoA and PCoA) were visible on CTA and did not measure the diameters, contrary to a study by Zhou et al. [26]. While some studies based on the analysis of the circle of Willis variants have not proven a

relationship between anatomy and stroke outcomes [33,34], our results are consistent with other earlier research [24,26,35]. We think the primary cause of this discrepancy between studies is that various classification systems were employed by the authors in the analysis of CoW arteries. These conflicting results show that additional research on the CoW anatomy is needed.

Patients in our three groups had comparable age and sex distribution; however, when it comes to stroke causes, the patients in the unfavorable collateral group had a higher incidence of arterial hypertension and atrial fibrillation. Chronic hypertension is a well-known risk factor for worse outcomes in stroke patients. According to a review by Noor et al., hypertension, peripheral artery disease, and younger age correlate with poor functional outcomes in patients undergoing EVT [36]. Furthermore, Alobaida et al. concluded that atrial fibrillation was also linked to a worse functional recovery but not with intracerebral hemorrhage or a higher death rate [37]. Worse outcomes of stroke connected to atrial fibrillation were found in a study by Rebello et al., where they explained that atheroembolic strokes appear to be associated with improved collateral flow in comparison to patients with atrial fibrillation-related strokes [38]. This was further supported by other studies [39,40].

Studies have shown that patients with higher collateral activation have lower NIHSS values on admission [41,42]. This is in accordance with our study, explaining that higher NIHSS reflects larger ischemic tissue volume [43,44], as confirmed in our results. The admission blood pressure is another crucial aspect. Research has revealed that higher blood pressure levels prior to endovascular treatment decreased favorable 3-month functional independence [43,44]. Furthermore, lower post-treatment blood pressure was associated with better outcomes [45–47]. Our patients in all three groups had comparable systolic blood pressure values on admission, so we were unable to correlate that.

In our study, we also evaluated the onset-to-door time in order to assess any variations in the time between symptom onset and groin puncture. Our findings indicated that our patients had comparable onset times, and there were no delays in receiving endovascular therapy for either group. This information is particularly significant, as our patients were referred from distant primary stroke centers, highlighting the effectiveness of our well-established protocols and trained healthcare professionals in managing acute stroke patients. Numerous published studies have explored the impact of treatment delays on the prognosis of stroke patients. For instance, Snyder et al. [48] demonstrated that patients who presented within 6 h of stroke onset and underwent mechanical thrombectomy achieved better outcomes. Additionally, it is important to consider whether patients are directly admitted to comprehensive stroke centers, as this can lead to a shorter time to recanalization and consequently better outcomes [49]. Furthermore, additional research has shown that patients admitted during off-hours may experience poorer functional outcomes due to treatment delays [50]. These findings emphasize the importance of timely and efficient management of stroke patients, which can significantly influence their overall prognosis and functional outcomes.

There was no significant difference in the type of mechanical thrombectomy devices used among the three collateral groups. The majority of patients underwent the procedure using aspiration catheters. This indicates that the choice of mechanical thrombectomy technique was not influenced by the collateral status. With regards to collateral status and functional outcomes, the favorable collateral group had a higher percentage of patients with favorable mRS scores (indicating better functional outcomes) and a lower mortality rate compared to the other groups. This suggests that better collateral status may be associated with improved stroke outcomes. There were no significant differences between the collateral groups in terms of unsuccessful placement of endovascular devices, successful recanalization, TICI scores, number of passes during thrombectomy, control CT area of ischemia, and the presence of subarachnoid or intracerebral hemorrhage. These factors did not appear to be strongly influenced by the collateral status.

Our study revealed distinct profiles of CT perfusion parameters and stroke outcomes among the three groups (unfavorable, intermediate, and favorable) based on collateral status. The favorable collateral group had the lowest volume of core infarct and penumbra on CT perfusion maps compared to the other groups. Additionally, the mismatch ratio, which indicates the extent of mismatch between core infarct and penumbra, was the highest in the favorable collateral group, suggesting better perfusion patterns. The potential recuperation ratio, which represents the likelihood of recovery, was lowest in patients with poorly developed collaterals and highest in patients with well-developed primary and secondary collaterals. This suggests that better collateral status may contribute to a higher potential for recuperation.

Several studies correlated the presence of collaterals with automatically calculated values on CT perfusion. Regarding penumbra, some studies found that pial collaterals directly affect the proportion of recoverable penumbra [21], which is in accordance with our results. In addition, we also demonstrated that, as several CT perfusion studies have found, collateral status is related to core volume [51–53]. Our study showed that patients with the best collateral activation tended to have a higher rate of small/lacunar infarcts as a final core volume in comparison with low-collateral patients.

A higher recanalization rate [9] is correlated with collateral status, according to certain research evaluating leptomeningeal collaterals; however, other studies have not reached this conclusion [2,54]. According to our research, there was no correlation between successful recanalization and the degree of the collateral system. We also collected information on first-pass recanalization since it is known to be a predictor of better outcomes [55–59]. Our groups of patients had similar first-pass recanalization rates, which means that all our patients had similar treatment conditions.

Each of these studies underscores the significance of collateral circulation as a crucial factor in predicting the efficacy of endovascular therapy. Our study aligns with previous research, but we provide a significant added value by demonstrating the combined yet independent effect of both primary and secondary collaterals on patient survival and recovery.

Our study has several limitations. Firstly, we focused on patients within the "early" time window, which may limit the generalizability of our findings to patients in later stages of stroke. Additionally, our total sample size was limited, which could impact the statistical power and generalizability of our results. Nonetheless, we mitigated these limitations by conducting thorough examinations of comorbidities, meticulously analyzing radiological data, and collecting comprehensive information on thrombectomy procedures. Another limitation of our study is that the relationship between collateral status and stroke etiology was not assessed as analyzed in recent meta-analysis [58,59]. Despite these limitations, our study provides valuable insights into the impact of collateral circulation on endovascular therapy outcomes, offering a unique perspective on the independent contributions of primary and secondary collaterals to patient survival and recovery.

## 5. Conclusions

In conclusion, all our patients had equal conditions for achieving a successful treatment effect with mechanical thrombectomy, including arrival time and technical aspects of treatment. However, they differed in the cause of the ischemic stroke itself and the status of collaterals.

Our study revealed a significant relationship between collateral system status and CT perfusion values on emergency CT scans. The analysis demonstrated that both primary and secondary collateral systems work synergistically, with the circle of Willis playing a more prominent role in occlusions involving the distal internal carotid artery compared to M1 occlusions. These findings suggest a potential relationship between collateral circulation and CT perfusion parameters.

Our study's findings, which demonstrated better functional outcomes and higher survival rates in patients with good collateral circulation, provide valuable assistance to radiologists in making decisions about endovascular treatment. This is particularly relevant

for patients who are located far from comprehensive stroke centers, where longer transfer times may be experienced, as well as for those who receive radiological examinations in centers without access to sophisticated analyses like CT perfusion. Additionally, our results may be useful in cases of wake-up strokes or strokes with unknown onset times. Further studies with larger patient cohorts have the potential to enhance and expand the current recommendations for mechanical thrombectomy. By carefully analyzing patient age, comorbidities, and collateral circulation, interventional neuroradiologists can make informed decisions about the use of invasive procedures such as mechanical thrombectomy.

**Author Contributions:** Conceptualization, S.S. and M.M.G.; methodology, S.L.K.; software, S.S.; validation, K.D., D.B.M., and S.L.K.; formal analysis, A.M.; investigation, S.S.; resources, M.Č.-V.; data curation, M.Č.-V.; writing—original draft preparation, S.S., M.M.G., and D.B.M.; writing—review and editing, K.D. and S.L.K.; visualization, A.M.; supervision, M.M.G. All authors have read and agreed to the published version of the manuscript.

**Funding:** This research received no external funding.

**Institutional Review Board Statement:** The study was conducted in accordance with the Declaration of Helsinki and approved by the Institutional Review Board (or Ethics Committee) of University hospital Split (protocol code 2181-147-01/06/LJ.Z.-23.02).

**Informed Consent Statement:** Patient consent was waived due to retrospective study.

**Data Availability Statement:** The datasets analyzed during the study are available from the corresponding author on reasonable request.

**Acknowledgments:** The authors thank to the Radiology and Neurology Department team included in the treatment of acute ischemic stroke patients at the University Hospital of Split.

**Conflicts of Interest:** The authors declare no conflicts of interest.

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
