# Peer review of "Communicating Arteries and Leptomeningeal Collaterals: A Synergistic but Independent Effect on Patient Outcomes after Stroke"

_2035-8377, doi:10.3390/neurolint16030046_

Round 1

Reviewer 1 Report

Comments and Suggestions for Authors

I have the following comments:

1. Did you take into account the size/diameter of the a. comm. ant. a post. or only whether it was visible on the CTA? Could you put a note about this in the Discussion?

2. For the sake of reproducibility, please indicate the manufacturer of the instruments and software.

3. Is the group marked number 1 really favourable? I believe that groups 1 and 3 are confused in text, correct?

4. The Conclusion states: "especially for patients who are far from the primary stroke center  (and thus experience longer transfer time to PSC) and who are radiologically examined  but without sophisticated analyses like CT perfusion". You were really referring to primary centres, which according to the classification only provide CT and TL, and not secondary centres, which also provide endovascular treatment?

5. What is the validity of the study results for practice. In conclusion, it should be commented at what stage and under what conditions the patient can be excluded from recanalization treatment on the basis of "unfavourable" collateral.

6. In literature citations, capitalization should be made consistent and journal requirements should be checked.

7. In Conclusion, I would suggest that the two collateral systems apply synergistically, the Willis circle being essential for "T" occlusions and isolated a.car. int occlusions, while less so for M1 occludions.

Comments on the Quality of English Language

Typos:

line 154 systolic

line 224 concluded

line 255 final

Reviewer 2 Report

Comments and Suggestions for Authors

This is a nice study of the importance of collateral circulation in acute ischemic stroke. It is nicely presented and well referenced with rigorous statistical analysis. Some minor points: "systolic" is misspelled on line 154 and "agemcorrelate" should be corrected on line 223. 

1. Outcome after ischemic stroke was favorably impacted by the integrity of the collateral circulation specifically the communicating arteries and leptomeningeal collaterals 2. The article demonstrates the benefit of the collateral circulation through their analysis of CTA findings. 3. The study reinforces the general feeling that the better the collateral circulation the better the outcome. 4. I have no other suggestions to their methodology or controls. 5. I concur with the conclusion that primary and secondary collaterals contribute to improved outcome 6. I view the references as appropriate 7. No other comments.
